# MARGINAL: An Automatic Classification of Variants in *BRCA1* and *BRCA2* Genes Using a Machine Learning Model

**DOI:** 10.3390/biom12111552

**Published:** 2022-10-24

**Authors:** Vasiliki Karalidou, Despoina Kalfakakou, Athanasios Papathanasiou, Florentia Fostira, George K. Matsopoulos

**Affiliations:** 1School of Electrical and Computer Engineering, National Technical University of Athens, 15780 Athens, Greece; 2Molecular Diagnostics Laboratory, INRaSTES, National Center for Scientific Research NCSR Demokritos, 15341 Athens, Greece

**Keywords:** genomics, *BRCA1/2* genes, machine learning, rare variant interpretation, ACMG-AMP guidelines, variant pathogenicity, germline, cancer

## Abstract

Implementation of next-generation sequencing (NGS) for the genetic analysis of hereditary diseases has resulted in a vast number of genetic variants identified daily, leading to inadequate variant interpretation and, consequently, a lack of useful clinical information for treatment decisions. Herein, we present MARGINAL 1.0.0, a machine learning (ML)-based software for the interpretation of rare *BRCA1* and *BRCA2* germline variants. MARGINAL software classifies variants into three categories, namely, (likely) pathogenic, of uncertain significance and (likely) benign, implementing the criteria established by the American College of Medical Genetics and Genomics and the Association for Molecular Pathology (ACMG-AMP). We first annotated *BRCA1* and *BRCA2* variants using various sources. Then, we automatically implemented the ACMG-AMP criteria, and we finally constructed the ML model for variant classification. To maximize accuracy, we compared the performance of eight different ML algorithms in a classification scheme based on a serial combination of two classifiers. The model showed high predictive abilities with maximum accuracy of 92% and 98%, recall of 92% and 98% and specificity of 90% and 98% for the first and second classifiers, respectively. Our results indicate that using a gene and disease-specific ML automated software for clinical variant evaluation can minimize conflicting interpretations.

## 1. Introduction

With the advent of next-generation sequencing (NGS), new genes related to genetic disorders have been identified, leading to an exponential increase of DNA sequence variants being detected. Variant interpretation and classification is a complex process and is based on data from various perspectives that can many times prove to be discordant with respect to the classification of a particular variant. Thus, the American College of Medical Genetics and Genomics and the American Molecular Pathology (ACMG-AMP) have proposed a set of criteria to weight variant evidence, which in combination enable variants to be classified into five classification tiers i.e., pathogenic (P), likely pathogenic (LP), variants of uncertain significance (VUS), likely benign (LB) and benign (B) [1].

*BRCA1* and *BRCA2* play a central role in DNA repair through homologous recombination. RAD51 is a key component in the repair of double-strand breaks which binds to both *BRCA1* and *BRCA2* for the recognition and repair of damage, respectively. Therefore, *BRCA1* and *BRCA2* function in a critical pathway responsible for genome integrity [2]. In the case of the presence of loss-of-function variants, *BRCA1* or *BRCA2* are inactivated, and therefore, DNA repair through homologous recombination cannot be completed. This results in the accumulation of genetic defects within the cells. Females that carry germline *BRCA1* or *BRCA2* pathogenic variants are at high lifetime risk of breast and ovarian cancer diagnoses, which can be 70% and 44%, respectively [3]. *BRCA1* and *BRCA2* are the most well-studied and established genes in hereditary breast and ovarian cancer, while pathogenic variants in these genes are inherited in an autosomal dominant manner [4]. DNA variants in *BRCA1* and *BRCA2* genes can be interpreted and classified applying the ACMG-AMP criteria.

More specifically, *BRCA1*- and *BRCA2*-associated hereditary breast and ovarian cancer (HBOC) is characterized by an increased risk for breast cancer and ovarian cancer (including fallopian tube and primary peritoneal cancers) and, to a lesser extent, for other cancers such as male breast, prostate and pancreatic cancer, seen mainly in individuals with a *BRCA2* pathogenic variant. Conferred cancer risks differ for pathogenic variants in *BRCA1* or *BRCA2* genes [5]. Having established that each gene is unique, the two best-studied genes, i.e., *BRCA1* and *BRCA2*, were selected for our study in order to increase the accuracy of our results. Precise clinical interpretation of identified variants in these genes can play a major role in providing the best therapeutic care to patients that are found to be carriers as well as in the early detection of inherited types of cancers.

Although the main purpose of ACMG-AMP guidelines is to enable reliable interpretation and classification of variants, application of the ACMG-AMP criteria is still subject to some discrepancies between individual interpreters. A possible reason for this is that the ACMG-AMP guidelines are generally limited to evaluating each criterion individually, but do not use specific algorithms for implementing these guidelines.

Furthermore, even though a variety of databases such as ClinVar and Exome Aggregation Consortium (ExAC) and in silico tools such as SIFT and PolyPhen-2 are available online, there is the need for an application that can combine all the data to provide one specific outcome for the classification of genetic variants. Notably, many of these databases often contain incorrectly classified variants or even contradictory records on the evaluation of pathogenicity, while gene and disease may determine the applicability and weight assigned to certain criteria. Thus, researchers should consider this before classifying specific gene variants and developing more focused guidance [1].

Many automatic tools that implement ACMG-AMP criteria and rules have been developed to address these challenges. For instance, InterVar is an automated tool that helps human reviewers to interpret the clinical significance of variants in any Mendelian gene [6], while Cancer SIGVAR contributes to the interpretation of hereditary cancer-associated germline variants [7], PathoMAN allows the automation of germline variant curation in clinical cancer genetics [8], CardioVAI enables variant interpretation in the diagnosis of cardiovascular diseases [9] and the GenOtoScope tool automatically classifies variants that may be associated to congenital hearing loss [10]. All these tools are based on the implementation of ACMG-AMP criteria, while the final classification of the variants depends on the number of verified criteria combined with the rules provided in the 2015 ACMG-AMP guidelines. In addition, there are other automated tools such as Xrare, which is a machine learning approach for prioritizing variants associated with disease using a set of phenotypic (based on phenotype-similarity measures) and genetic (based on ACMG evidence, in silico computation scores and population-level scores) features [11]. Using a gradient-boosting algorithm all these genotypic and phenotypic features are combined to predict the pathogenicity of a variant. Another machine learning-based tool, RENOVO, uses publicly available data to reclassify 67% (with an estimated precision of > 90% using a random forest classifier) of germline variants of unknown significance as pathogenic or benign and generates a pathogenicity likelihood score (PLS). It uses pre-processed features obtained from ClinVar and Annovar to cover the highest number of the ACMG-AMP guidelines (functional scores such as MutPred, Mutation Assessor, SIFT and conservation scores such as GERP++_RS) [12]. On the other hand, LEAP is a machine-learning method that focuses on classifying missense variants in genes related to hereditary cancer and cardiovascular disease. The method uses features that are grouped in categories derived from the ACMG-AMP guidelines for variant interpretation (such as functional prediction, splice prediction, evolutionary conservation and health history). Based on these categories, the contribution of evidence categories to model performance can be compared [13]. Furthermore, VarCall XT (a multifactorial extension of VarCall), is a Bayesian statistical model to analyze the likelihood of pathogenicity for individual missense variants in *BRCA1* and *BRCA2* in relation to HBOC, as well as to examine the predictive accuracy resulting from adding in silico or family-based data to the functional data (the original function-based VarCall model achieved 98% accuracy for classifying *BRCA1* and *BRCA2* VUS, which was the top model based on overall accuracy for both genes) [14]. Finally, machine learning models developed to interpret DNA variants in three MODY (maturity-onset diabetes of the young) genes, i.e., *HNF1A*, *HNF4A* and *GCK*, using ACMG-AMP criteria revealed the necessity of applying different weights according to the MODY genes to ensure accuracy in functional classification (overall accuracy above 95% using logistic regression) [15]. Another approach in the literature combines artificial intelligence and the use of structure features. More specifically, an important component of this study was to establish a useful statistical deleteriousness prediction system for proteins or peptides based on a computational method for predicting the activity of p53 variants using structure features [16].

Herein, we present MARGINAL (autoMatic clAssification of bRca Genes usIng machiNe leArning modeL), a software that includes a machine learning model to support variant classification in *BRCA1* and *BRCA2* genes based on ACMG-AMP guidelines. To the best of our knowledge, this is the first approach that combines the implementation of 17 ACMG-AMP criteria and a machine learning model that uses these criteria as features for automated classification of germline rare variants in *BRCA1* and *BRCA2* as P/LP, VUS and B/LB variants. All variant classification tools that include a machine learning model used to date use different available scores directly as features, in accordance with the ACMG-AMP guidelines and for different classification purposes.

## 2. Materials and Methods

### 2.1. Data Acquisition

DNA variants of *BRCA1* and *BRCA2* genes were acquired from CanVaS (a Cancer Variation reSource) database. The data set comprises data from germline genetic testing of cancer patients and integrates functionally annotated rare variants in established or suspected cancer susceptibility genes [17]. The variants studied all derived from patient cohorts, i.e., high-risk individuals and/or individuals diagnosed with female breast cancer, male breast cancer, ovarian cancer and pancreatic cancer. The required input for the first stage of MARGINAL software is a simple text file including a list of variants that will then be annotated with a set of required information by the Ensembl Variant Effect Predictor (VEP) [18], which is the main annotation tool; 497 unique rare variants of the original total number of 504 variants were annotated via VEP (215 and 282 variants reported in *BRCA1* and *BRCA2*, respectively). Among these, missense variants accounted for 40%, while frameshift, synonymous and stop-gain variants accounted for 26%, 17% and 14%, respectively. A small proportion involved in-frame deletions/insertions and splice variants. VEP compares input variants to known variants from the Ensembl variation database in order to extract annotations (very large copy number variants—CNVs—are not available). Therefore, for both genes, the resulting data set includes 173 P/LP variants, 194 B/LB variants and 130 VUS (the data were previously classified as P, LP, VUS, LB or B using CanVaS). These classifications were thus used in machine learning modeling to train and test algorithms for the final variant classification (to generate model training labels, P and LP variants were grouped and will be called “P/LP”, while B and LB variants were grouped and will be called “B/LB”). Ultimately, this results in three classes—P/LP, VUS and B/LB variants—for the final variant classification used in machine learning modeling. Therefore, we reduced the classification complexity of the machine learning model from five to three classes, thereby increasing the separation power and classification accuracy.

### 2.2. Methodological Scheme

MARGINAL software consists of three main steps: (1) Variant annotation using several annotation tools (VEP, MMSplice and RepeatMasker) and databases (ClinVar and CanVaS) to obtain computational predictive data, population frequency data and genomic annotation, (2) ACMG-AMP criteria implementation and their automated coding at values 0 or 1 and (3) Machine Learning modeling using the ACMG-AMP criteria as feature vectors for the final classification of the variants (P/LP, VUS and B/LB). Algorithm implementation and data analysis were performed with Python3 (packages/functions from Scikit-Learn library v.1.0.2 [19] were used for machine learning modeling).

#### 2.2.1. Variant Annotation

The variant annotation is based on Genome Reference Consortium Human Genome Build 37 (GRCh37). Various annotation resources were used for automated interpretation, including (1) populational data derived from the Exome Aggregation Consortium (ExAC) and the Genome Aggregation Database (gnomAD), (2) predicted data from dbNSFP (v4.2a) [20,21] and dbscSNV (1.1) [22] and (3) other available databases. In particular, by utilizing the web interface provided by Ensemble Variant Effect Predictor (VEP) [18], we were able to combine the annotation information that was manually extracted with those of other annotation sources, MMSplice [23], RepeatMasker from the UCSC Genome Browser [24,25], ClinVar [26] and CanVaS databases. All required information for *BRCA1* and *BRCA2* genes was extracted and processed from these sources and was subsequently automatically integrated into a CSV file.

#### 2.2.2. ACMG-AMP Criteria Implementation

MARGINAL software implements 17 out of 28 ACMG-AMP criteria automatically. In particular, according to variant annotation, it can automatically generate predictions on ten and seven criteria that conclude a pathogenic and a benign variant classification, respectively (Table 1). If a criterion is positive, the algorithm will assign 1, otherwise, it will assign 0. The rest of the ACMG-AMP criteria are based on categories that are specific to each case (such as familial co-segregation data or de novo status and allelic data) and require user input. Thus, these were not included in the proposed process.

PVS1 criterion suggests strong pathogenicity and is recommended for loss-of-function (LOF) variants (represented as frameshift indel, stop-gain, stop-loss and splicing variants in canonical transcripts). LOF is a known mechanism of disease in *BRCA1* and *BRCA2* genes. Considering that the vast majority of LOF variants in these genes are considered pathogenic, with the exception of those located in the last amino acids of the last exon of *BRCA2*, PVS1 is assigned as 1 for all these variants; for all other variants, PVS1 is assigned a value equal to 0.

For the implementation of PS1, PM5, PP5, and BP6 criteria, as a reference to identify pathogenic and benign variants, we interrogated ClinVar, excluding variants interpreted from a single submitter and variants with conflicting interpretations (review status ≥ 2). More specifically, when a missense variant is pathogenic, then if a different nucleotide change resulting in the same amino acid change is known to be pathogenic, PS1 is assigned a 1. In contrast, if a new missense amino acid change occurs at the same position as another pathogenic missense change, then PM5 is assigned a 1. We used the annotation information derived from VEP related to CDS_position (relative position of base pair in coding sequence), protein_position (relative position of amino acid in protein), amino_acids (amino acid change in case the variant affects the protein-coding sequence) and GIVEN_REF (reference allele from input) fields [18]. To ensure that these changes do not impact splicing, we inferred the “pathogenicity” score from MMSplice for missense variants (pathogenicity score < 0.85). Concerning PP5 and BP6 criteria, if a variant has recently been identified as pathogenic by a reputable source, but an independent evaluation cannot be performed, then PP5 is applied. If a variant has recently been identified as benign by a reputable source, but an independent evaluation is not possible, then BP6 is applied. As a result, if a variant is identified as “pathogenic” or “likely pathogenic” by ClinVar, PP5 is assigned as 1, otherwise, if a variant is identified as “benign” or “likely benign” by ClinVar, BP6 is assigned as 1. As a result of the limitation mentioned above, i.e., review status ≥ 2, we can consider the PP5 and BP6 criteria to approach PS3 and BS3, respectively, which is to say that they can be considered to be functional evidence [8].

BA1, BS1 and PM2 criteria that are related to population frequency were implemented based on ExAC [27] and gnomAD databases [28] and more specifically the non-TCGA and non-cancer data sets, respectively, considering all populations. If the variant has an allele frequency > 5%, BA1 is assigned as 1. If the allele frequency in a control population is higher than expected for the disorder (in our case, the threshold for rare variants is 1%), BS1 is assigned as 1. For dominant inheritance, as in the case of *BRCA1* and *BRCA2*, when a variant is absent in all control subjects from the above databases, PM2 is assigned as 1, as the rarity of the variant advocates for pathogenicity.

When the prevalence of the variant in affected individuals is significantly higher than in controls, then PS4 is applied. As a result of a case-control study (between the CanVaS population of interest and gnomAD exome control database), all variants whose odds ratio is higher than 2 and *p*-value is less than 0.05 for cancer risk are coded as 1 for PS4. We inferred the odds ratio and *p*-value from Fisher’s exact test.

If a protein’s length changes due to in-frame insertions or deletions in a non-repeat region or stop-loss variants, PM4 is applied. Our annotation for the repeat region was based on the ‘‘rmsk’’ database from the UCSC Genome Browser (using the UCSC Table Browser data retrieval tool) [24,25]. The database records the results of RepeatMasker, which were generated by screening DNA sequences for repeats. If the variants are in-frame insertions or deletions in the non-repeat region, or stop-loss variants, PM4 is assigned as 1. If the variants are in-frame insertions or deletions in the repeat region, BP3 is assigned as 1.

PM1 criterion includes moderate evidence of pathogenicity if specific protein domains are thought to be essential for protein function and/or mutational hot spots and all missense variants in these domains are known to be pathogenic. In addition, benign variants are not located in these domains. For *BRCA1* and *BRCA2*, there are critical and well-established functional domains, but not without benign variants. Therefore, PM1 is assigned as 0.

If a missense variant occurs in a gene in which missense variants often result in disease and which tends to have fewer benign missense variants, PP2 is applied. However, for a missense variant in a gene where truncating variants are the predominant cause of disease, BP1 is applied. Since there are not insufficient data for *BRCA1* and *BRCA2* concerning these criteria, PP2 and BP1 are assigned as 0.

If multiple lines of computational evidence point to a deleterious effect of a variant on the gene or gene product (conservation, evolutionary, splicing impact, etc.), then the supporting pathogenic evidence of PP3 is assigned as 1. However, if multiple lines of computational evidence suggest no impact on the gene or gene product, BP4 is assigned as 1 for supporting benign evidence. Considering that all in silico programs agree on the prediction, this evidence can be considered supporting, and PP3 or BP4 is applied. The computational data can be obtained via VEP from the “dbNSFP” database, using MetaSVM and Condel predictions [29] to predict deleteriousness and the GERP++ score (GERP++_RS) to predict evolutionary conservation. The impact of splicing can be assessed using VEP from the “dbscSNV” database, using adaptive boosting and random forest scores (ada_score and rf_score, respectively). If at least one of the above in silico data exists and MetaSVM and Condel predictions indicate a deleterious variant, GERP++_RS is higher than 2, ada_score and rf_score are higher than 0.6 and PP3 is assigned as 1, otherwise, BP4 is assigned as 1.

In the case of a synonymous variant, known not to alter splicing and the nucleotide position not being highly conserved, BP7 is applied. By using VEP, the effect on splicing can be predicted from the ‘‘dbscSNV’’ database and the conservation information can be inferred from the “dbNSFP” database as above. If ada_score and rf_score are less than 0.6 and GERP++_RS is less than 2, BP7 is assigned as 1.

#### 2.2.3. Machine Learning Modeling

In this study, a machine learning model is proposed in order to discriminate between P/LP, VUS and B/LB variants. After a detailed analysis, we concluded that the optimal classification result is achieved via two two-tier serial classifiers. We compared the performance of eight machine learning algorithms, namely, logistic regression (LR), linear discriminant analysis (LDA), k-nearest neighbors classifier (KNN), decision tree classifier (CART), naive Bayes (BernoulliNB), support vector machine (SVM), random forest (RF), and multi-layer perceptron (MLP) classifier, in a classification scheme based on a serial combination of two classifiers. In particular, instead of a typical three-tier classifier, we propose two serial classifiers to reduce the three-tier to two-tier classification to simplify the machine learning model and thus increase separation power. In this way, as illustrated in Figure 1, the same feature vectors were used for both classifiers. The first classifier was trained for separating VUS from all other variants (B/LB and P/LP, labeled as “other variants”) and the second classifier was trained to discriminate between the remaining two classes—B/LB and P/LP.

Machine learning algorithms were trained based on the scikit-learn library in Python on version 1.0.2 [19]. The training set consisted of 80% of the total variants, while the test set consisted of the remaining 20% (no other criteria were applied to select variants). The different models were trained using three-fold cross-validation to evaluate and compare their performances, and this procedure was repeated 20 times. To determine the best approach, we used accuracy, F1 measure, precision, recall and specificity. When implementing algorithms, the default scikit-learn library parameters were used unless otherwise specified.

In addition, an optimization step was performed on the parameters of each algorithm to improve their classification performance. The parameter selected for LR was solver = ‘newton-cg’, for BernoulliNB was fit_prior = none, for SVM was kernel = ‘linear’; the parameters for RF were n_estimators = 200, max_features = 11 and for MLP were hidden_layer_sizes = 11, solver = ‘lbfgs’ and max_iter = 400.

To evaluate the significance of the features used to train the model, we applied feature importance analysis with SHAP (Shapley additive explanations) values [30] on the training set, using the SHAP Python library. This method allows us to determine precisely how the features contribute to the model output. Furthermore, we computed Spearman’s correlation [31] among features to examine feature collinearity. As a result of our analysis based on the CanVaS database, we found that RF (nonlinear machine learning classification algorithm) or MLP (fully connected class of feedforward artificial neural network) are the best options for classifier 1, and LR (linear) or BernoulliNB (a variant of naive Bayes, a classification algorithm of machine learning based on Bayes theorem) for classifier 2. The flow diagram of the whole process is presented in Figure 1.

## 3. Results

### 3.1. Machine Learning Model Comparison

We initially implemented the machine learning (ML) algorithms using the full set of 17 features (Table 1) for training and test sets and trained them to distinguish P/LP, VUS and B/LB variants. Variants from the CanVaS database are used as reference. CanVaS is a national database that records rare germline genetic variation of Greek cancer patients. This information is accompanied by relevant phenotypic and segregation data, enabling accurate variant classification. Each unique variant in CanVaS has been individually assessed and classified accordingly by our group, following the ACMG guidelines, rendering this data set a valuable source for the development of in silico tools in need of accurately labeled data. To detect the collinearity among feature vectors, we computed Spearman’s correlation for the ACMG-AMP criteria (Figure 2). According to the results of the study, no correlation coefficient greater than or equal to 0.85 was detected. Consequently, none of the ACMG-AMP criteria needed to be excluded. Based on the machine learning modeling, we compared the performance of eight ML algorithms in a serial combination scheme of two classifiers, as mentioned above, after three-fold cross-validation on the training set. General features of the original data set that was used for training and testing the two classifiers are shown in Table 2.

Following the analysis of the test set, classification results of the algorithms are provided in Table 3. In Figure 3, the three ML algorithms with the best accuracy are presented (RF, MLP and BernoulliNB for classifier 1 and LR and BernoulliNB for classifier 2). The best overall accuracy achieved for the first classifier was 92% (true negatives = 64, false positives = 8, true positives = 28 and false negatives = 0 for the RF and MLP classifier and true negatives = 67, false positives = 5, true positives = 25 and false negatives = 3 for the BernoulliNB), while for the second it was 99% (true negatives = 36, false positives = 0, true positives = 37 and false negatives = 1 for LR and BernoulliNB classifier) for *BRCA1* and *BRCA2* variants. In Table 3, additional performance metrics demonstrate the machine learning model’s performance (for classifier 1 recall = 0.92, specificity = 0.96 and F1 score = 0.92 and for classifier 2 recall = 0.99, specificity = 0.99 and F1 score = 0.99). Based on receiver operating characteristic (ROC) analysis (Figure 3d,e), there was a slight difference between the ROC curves and a fairly high area under the ROC curve (AUROC) of 0.92 for the RF and MLP and 0.96 for the BernoulliNB as the first classifier, while it was over 0.99 for the LR and BernoulliNB as the second classifier, indicating that most variants would have been correctly classified using MARGINAL software. The misclassified variants, i.e., the variants whose class prediction is different from the actual class (label), predicted using eight ML algorithms for classifiers 1 and 2 based on the test set provided by the CanVaS database are presented in Appendix A. The total number of final labels predicted via the best ML algorithms for each class based on the CanVaS database and the total number of labels provided by the ClinVar database are shown in Appendix A.

By choosing RF or MLP for classifier 1 and LR or BernoulliNB for classifier 2, performance metrics for each class are shown in Table 4. These results indicate that our ML model has excellent performance characteristics and shows the ability to discriminate between the two classes for each of the two high-scoring classifiers (while RENOVO achieved a training set accuracy of 99% on established pathogenic/benign variants in ClinVar and a test set accuracy of 95% on variants whose interpretation has changed over time, in addition to showing an estimated precision greater than 90% in comparison to ClinVar VUS as pathogenic or benign [12]). It is important to note that this comparison is only for general reference and is not intended to be a direct comparison of performance, since the two tools differ in terms of objectives and design.

In addition to the final model of the two serial classifiers, we first independently studied each of the eight ML algorithms described above as a typical three-tier machine learning model to discriminate between P/LP, VUS and B/LB variants, using the same features. The whole machine learning procedure was the same as the one followed above. Regarding the test set, the classification results of the algorithms, after the testing of each of them independently, are provided in Table 5. According to our research, we found that independently tested algorithms are highly accurate in predicting hundreds of variants in *BRCA1* and *BRCA2* genes. Therefore, as we can see, there is only a slight difference between the algorithms’ scores, and the best accuracy score was 92% on the test set. The additional performance metrics in Table 5 also highlight the good quality of the algorithms’ predictions (recall = 0.93, specificity = 0.96 and F1 score = 0.92 for the CART, RF and MLP classifier).

### 3.2. Feature Selection

As described above, in order to compute feature importance in the machine learning model and study a possible reduction of initial features, we randomly selected RF for the first classifier and performed feature importance analysis with SHAP values on the training set. The values for the most important features are shown in Figure 4. Accordingly, the 17 ACMG-AMP criteria are ranked hierarchically as follows: BP6, PVS1, PP5, PM2, PP3, BP4, PS4, PM4, PS1, PM5, BS1 and BP7. The last five features are constant and equal to zero, so their values are not shown in the figure. The results of our study revealed that these five features, i.e., PP2, PM1, BP3, BA1 and BP1, with average SHAP values ≤ 0.01, could be excluded but without performance improvement. Therefore, we kept the initial total of 17 features.

### 3.3. Performance Evaluation on ClinVar Data Set

Next, we evaluated the performance of our method by choosing the three ML algorithms with the best accuracy (RF, MLP and BernoulliNB for classifier 1 and LR, BernoulliNB and LDA for classifier 2) for variants that do not exist in the initial data set (CanVaS), using the ClinVar data set. Overall, there are 11932 unique variants reported in *BRCA1* and *BRCA2* (excluding variants that already exist in CanVaS, are interpreted from a single submitter, have conflicting interpretations, or variants for which interpretations are not provided), which comprised the test set. The interpretations from ClinVar were used in machine learning modeling to test algorithms for the final classification of variants after having been trained on the CanVaS data set according to the procedure described above. Furthermore, we excluded PP5 and BP6 criteria concerning ClinVar interpretation in order to avoid a potential bias in the classifier training. As a result of the analysis of the test set, the best overall accuracy achieved for the first classifier was 91% (for the BernoulliNB classifier), while for the second it was 98% (for the LDA classifier). Figure 5 and Table 6 show additional performance metrics, and the results of ROC analysis are shown in Figure 5e,f. Performance metrics for each class are shown in Table 7. Based on the results of the analysis with the ClinVar data set, MARGINAL software enables the prediction of clinical significance of *BRCA1* and *BRCA2* variants, a particularly useful feature that can find wide application in clinical practice. In the case of the CanVaS database, this combination of classifiers also has excellent performance characteristics (maximum accuracy of 92% and 98%, recall of 92% and 98% and specificity of 90% and 98% for the first and second classifiers, respectively), which slightly differ from those of the initial optimal combination—RF or MLP for classifier 1 and LR or BernoulliNB for classifier 2. Consequently, for our final machine learning model, we chose the last combination—BernoulliNB for classifier 1 and LDA for classifier 2—as it outperformed the other algorithms in the analysis based on the ClinVar database.

## 4. Discussion

As a result of advances in next-generation sequencing (NGS) technology, clinical laboratories perform a greater number of genetic tests investigating a number of genes, for the elucidation of genetic predisposition. New challenges in sequence interpretation have therefore emerged, and numerous new guidelines have been introduced regarding the clinical interpretation and reporting of sequence variants. Supporting evidence for variant classification can be obtained using a variety of computational tools based on different algorithms and databases, including SIFT [32], MutationTaster [33] and GERP++ [34], among others. Nevertheless, all of these computational tools evaluate genes using the same rules, whereas the accuracy of interpretation may vary considerably depending on the gene, protein sequence and functional domain. Moreover, a variety of software programs should be used for sequence variant interpretation since different algorithms provide different advantages and disadvantages. In this study, in light of the need to assess variant pathogenicity in each gene individually, we present MARGINAL, a software that combines the implementation of 17 ACMG-AMP criteria with a machine learning model to improve rare variant interpretation in *BRCA1* and *BRCA2*, while minimizing conflicting interpretations. MARGINAL software consists of three main steps: (1) variant annotation by combining various annotation tools, (2) ACMG-AMP criteria implementation and (3) machine learning modeling for the final variant classification (pathogenic or likely pathogenic (P/LP), variants of uncertain significance (VUS) and benign or likely benign (B/LB)).

It should be noted that according to the ACMG-AMP guidelines, 28 criteria were equally weighted using the proposed rules of variant classification. There is, however, a possibility that different types of criteria may contribute differently to determining pathogenicity. Thus, in our study, we used a machine learning approach in place of these classification rules to more accurately predict and quantify gene variant pathogenicity. There are several automated tools that use machine learning models to classify variants, however, each of them implements different features. In particular, most tools are limited to using the required predicted scores directly, strictly following the ACMG-AMP guidelines (without implementing the relevant ACMG-AMP criteria), or using different categories of evidence such as data deriving from functional assays combined with in silico protein predictors and segregation data [12,13,14]. To the best of our knowledge, the only study using ACMG-AMP criteria as features focused on the necessity of applying different weights for the ACMG-AMP criteria in relation to different MODY genes, as a means to accurately interpret genetic variants causing maturity-onset diabetes in the young. All known tools are also suggested for the classification of different variants (found in other genes) and for different clinical purposes. In fact, few of them focus on variants of specific genes or diseases. On the contrary, MARGINAL is the first software, to our knowledge, that implements the ACMG-AMP criteria and uses them as features for the targeted classification of rare variants in *BRCA1* and *BRCA2*. Ultimately, all of the existing tools address different aspects of the variant classification process, but they are not identical to MARGINAL software and therefore cannot directly be compared. An important aspect that should be noted is that models published in the literature are often trained using public databases of variants whose clinical interpretation may change over time (e.g., ClinVar database), thus leading to inaccurate predictions. Consequently, the lack of standardization and consistency in classifications renders pathogenicity predictors ineffective. On the contrary, the classification of variants in *BRCA1* and *BRCA2* resulting from MARGINAL software is based on the CanVaS database, which is a population-specific database. The clinical interpretation of variants listed in the CanVaS database has been carried out over time, incorporating segregation and functional data, while including many recurrent or founder variants, resulting in more reliable and accurate classifications.

Another issue we wish to emphasize is that the variant data set that has been used (CanVaS) refers to GRCh37 (hg19) assembly and includes transcripts of NCBI’s reference sequence (RefSeq) database. Consequently, our study evaluates the impact of the variants in *BRCA1* and *BRCA2* genes on transcripts NM_007294 and NM_000059, respectively, which are predominantly used worldwide by experts. Therefore, corresponding options must be made for the extraction of the annotation information provided by VEP.

Considering variant classification, we concluded with two serial classifiers, as described above. Numerous studies have shown that this proposed approach to a serial combination of classifiers aims to improve the performance of unstable or weak classifiers [35]. Consequently, although we started with a typical three-tier machine learning model, we proceeded to the alternative model of two serial classifiers in order to improve the classification performance. Our results certainly show an increased enhancement in the accuracy as well as in additional performance metrics (92% accuracy, 92% recall (sensitivity) and 90% specificity for the naive Bayes as the first classifier for separating VUS from all other variants and 98% accuracy, 98% recall and 98% specificity for the linear discriminant analysis as the second classifier for separating B/LB from P/LP variants). It is important to note that before we came to this specific combination of classifiers, we evaluated the other two combinations of class separation: a) other variants—B/LB and P/LP–VUS and b) other variants—P/LP and B/LB–VUS. Comparing the accuracy scores, we concluded that the optimal combination was the one we chose.

A limitation of this study involves the selection of variants, which is based on a national database, i.e., CanVaS, which includes genomic data from Greek individuals only. As previously reported, the Greek population is influenced by strong founder effects, and therefore, many of the variants tested can be population-specific [36,37,38,39]. Although the variant classification per se described herein might not be applicable to genetic data deriving from multi-ethnic cohorts, the approach to automatically classify variants with unknown significance can find broad application. It should be noted, though, that many of the ACMG-AMP criteria (e.g., the criteria based on computational, predictive and functional data, among others) are “population-agnostic”, meaning that they are not influenced by the presence of the variant in a population.

Some future extensions could be considered for MARGINAL software to improve the efficiency and accuracy of variant classification. Gaining access to additional co-segregation and functional data for *BRCA1* and *BRCA2* will enable the application of further ACMG-AMP criteria, an important step for more accurate classification. Moreover, MARGINAL software could be modified with the appropriate adjustments for a future study combining the use of structure features to see how this approach would influence the classification results. Finally, the technique implemented herein could be extended to other genes by providing a general framework for interpreting variants throughout the genome and thus increasing overall statistical power. Despite its complexity and evolution, variant classification remains an important field that requires continuous improvement and development of multiple computational tools. In clinical practice, MARGINAL software is expected to enhance the accuracy of gene variant interpretations and contribute to more personalized patient care.

## 5. Conclusions

Despite the substantial progress towards achieving accurate clinical diagnostics in recent years, there are still numerous obstacles we need to overcome in order to provide the best patient care. MARGINAL uses a machine learning model that was trained on well-annotated clinical data (CanVaS database), making it a powerful tool for the interpretation of disease-associated genetic variation. Considering its high performance and accuracy in the classification of variants in *BRCA1* and *BRCA2*, we believe that it could provide sufficient evidence for the localization of pathogenicity in these specific genes and thus contribute to the prognosis and/or diagnosis of breast or ovarian cancer. As a result, by enhancing prediction rates and ensuring precise genetic consultations, patient survival can be significantly improved.

## Figures and Tables

**Figure 1 biomolecules-12-01552-f001:**
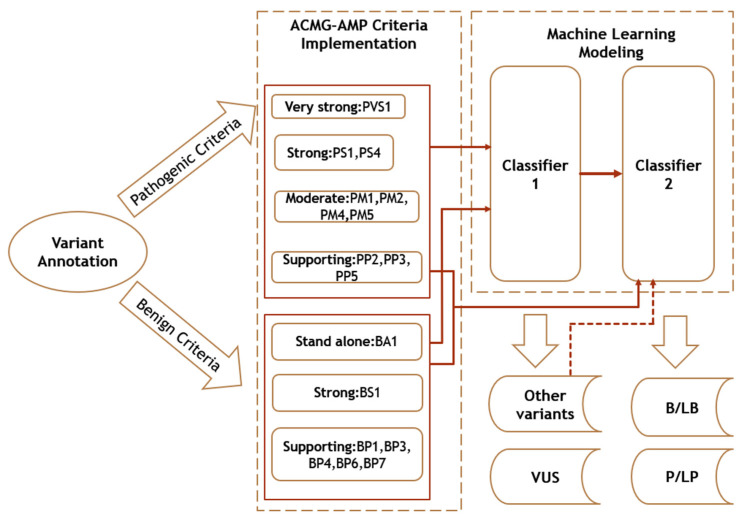
Schematic overview of the proposed method. The steps of the method are presented: (1) variant annotation, (2) ACMG-AMP criteria implementation and (3) machine learning modeling of two serial classifiers (the first classifier discriminates the variants of uncertain significance (VUS) from the rest variants, denoted as “other variants”, and the second classifier discriminates the benign or likely benign (B/LB) from the pathogenic or likely pathogenic (P/LP) variants).

**Figure 2 biomolecules-12-01552-f002:**
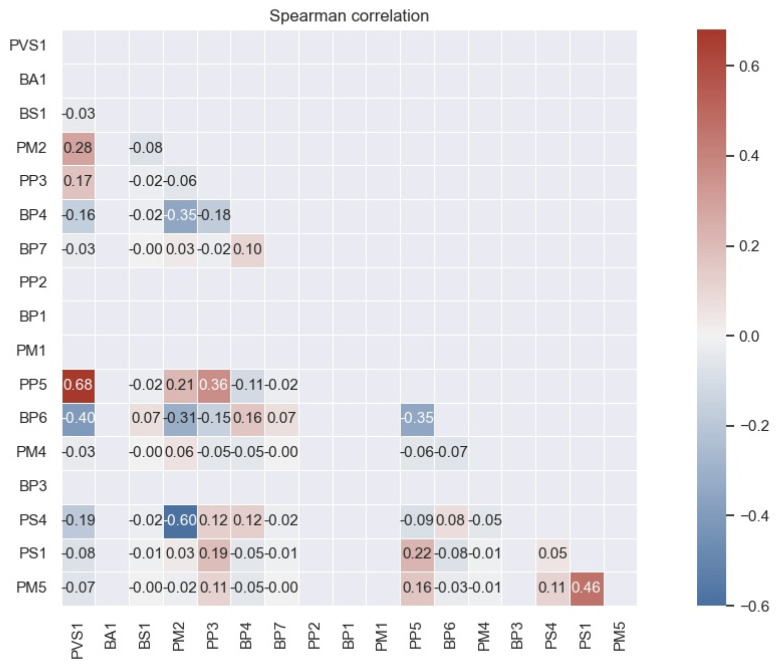
Spearman correlation among feature vectors. Darker shades identify higher correlation in terms of absolute values.

**Figure 3 biomolecules-12-01552-f003:**
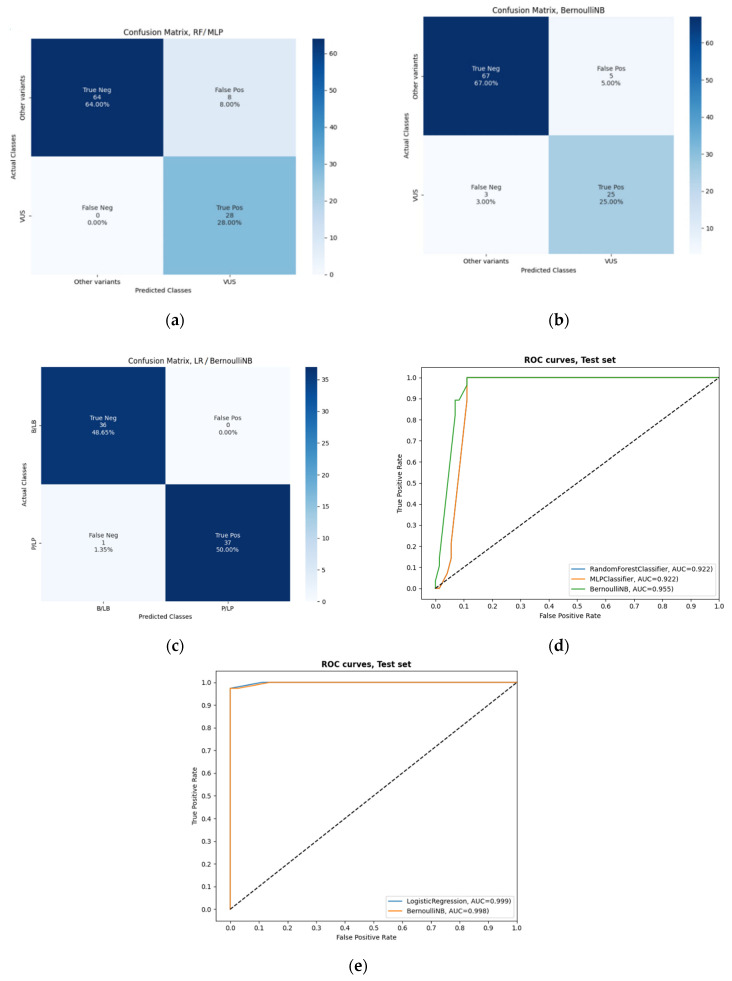
Classification results of the three best machine learning (ML) algorithms for classifier 1 and the two best ML algorithms for classifier 2 on the test set: (**a**) confusion matrix generated by implementing random forest (RF) or multi-layer perceptron (MLP) for classifier 1; (**b**) confusion matrix generated by implementing naive Bayes (BernoulliNB) for classifier 1; (**c**) confusion matrix generated by implementing logistic regression (LR) or BernoulliNB for classifier 2; (**d**) receiver operating characteristic (ROC) analysis—ROC curves and the values of the area under the ROC (AUROCs) to evaluate the performances for classifier 1; (**e**) ROC analysis—ROC curves and the values of the AUROCs to evaluate the performances for classifier 2.

**Figure 4 biomolecules-12-01552-f004:**
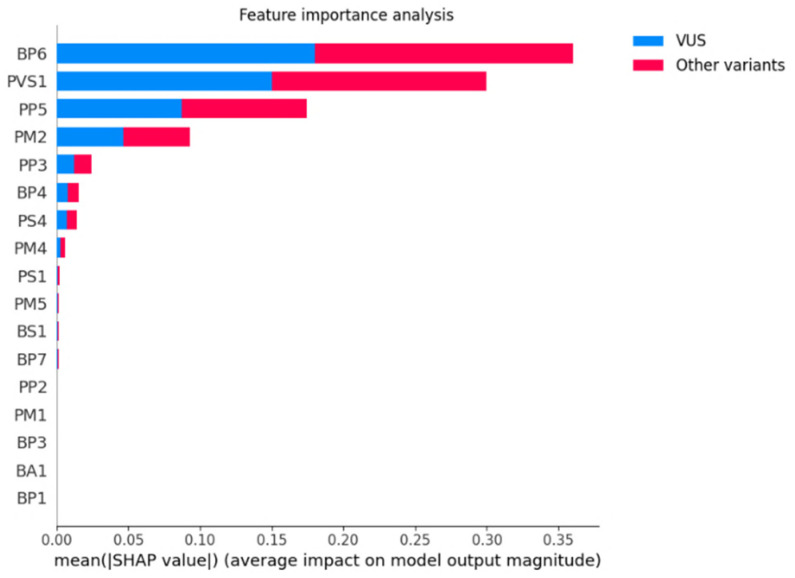
Ranking of feature importance using mean absolute SHAP values for the RF as classifier 1.

**Figure 5 biomolecules-12-01552-f005:**
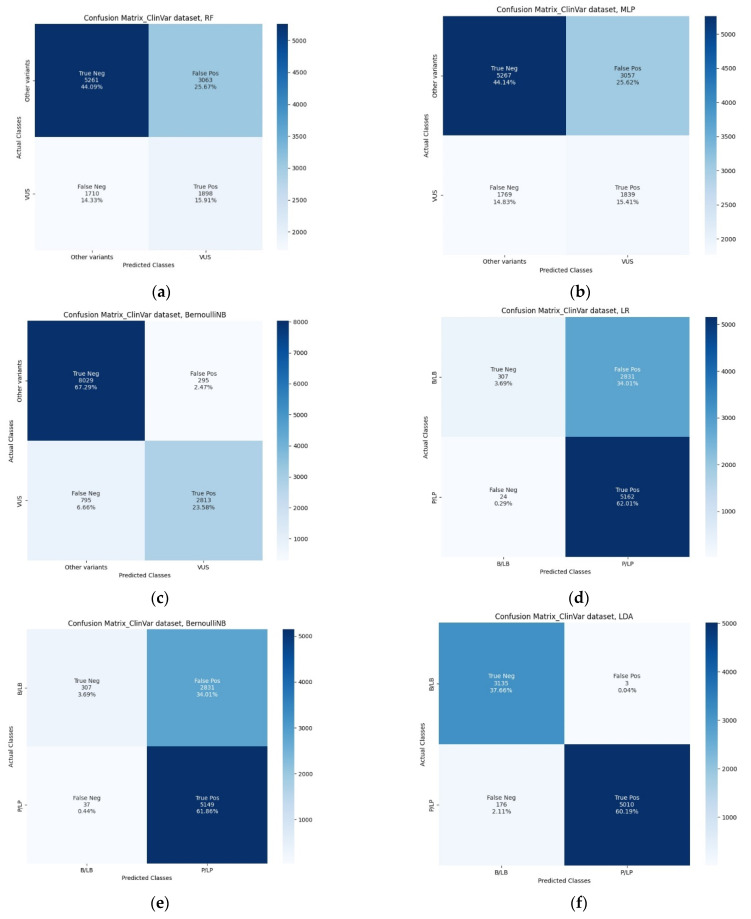
Classification results of RF, MLP and BernoulliNB for classifier 1 and LR, BernoulliNB and LDA for classifier 2 on the test set, based on the ClinVar data set: (**a**) confusion matrix generated by implementing RF for classifier 1; (**b**) confusion matrix generated by implementing MLP for classifier 1; (**c**) confusion matrix generated by implementing BernoulliNB for classifier 1; (**d**) confusion matrix generated by implementing LR for classifier 2; (**e**) confusion matrix generated by implementing BernoulliNB for classifier 2; (**f**) confusion matrix generated by implementing LDA for classifier 2; (**g**) ROC analysis—ROC curves and the values of the AUROCs to evaluate the performances of RF, MLP and BernoulliNB for classifier 1; (**h**) ROC analysis—ROC curves and the values of the AUROCs to evaluate the performances of LR, BernoulliNB and LDA for classifier 2.

**Table 1 biomolecules-12-01552-t001:** A total of 17 out of 28 American College of Medical Genetics and Genomics and the Association for Molecular Pathology (ACMG-AMP) criteria were automatically implemented.

ACMG-AMP Criteria	Category
PVS1	A null variant in a gene where loss-of-function is a known mechanism of disease
PS1	Same amino acid change but different pathogenic variant
PS4	Prevalence in affected individuals is significantly higher than controls
PM1	Located in functional domain/mutational hot spot without benign variants
PM2	Absent in all control subjects from population databases
PM4	In-frame insertion or deletion in non-repeat regions or stop-loss variants
PM5	New missense amino acid change occurs at the same position as another pathogenic missense
PP2	Missense variant in gene with a low rate of benign missense variants
PP3	Multiple computational studies point to a deleterious effect
PP5	A reputable source recently identifies a variant as pathogenic
BA1	Allele frequency >5%
BS1	Allele frequency in a control population is higher than expected for the disorder
BP1	Missense variant in a gene where truncating variants are the predominant cause of disease
BP3	In-frame indels in a repeat region
BP4	Multiple computational studies suggest no impact on gene or gene product
BP6	A reputable source recently identifies a variant as benign
BP7	A synonymous variant that does not alter splicing

**Table 2 biomolecules-12-01552-t002:** General features of the original data set that was used for training and testing the classifiers.

Classifier	Classes	Total Variants
1	Other variants	367
	VUS	130
2	B/LB	194
	P/LP	173

**Table 3 biomolecules-12-01552-t003:** Analysis of classification results of eight ML algorithms for classifier 1 and classifier 2—LR, linear discriminant analysis (LDA), k-nearest neighbors classifier (KNN), decision tree classifier (CART), BernoulliNB, support vector machine (SVM), RF and MLP classifier. The precision, recall score and f1-score are for the average of the classes (other variants—VUS and B/LB–P/LP) on the test set. The best ML algorithms are marked in bold.

Classifier	ML Algorithms	Precision	Recall	Specificity	F1-Score	Support	Accuracy
	LR	0.93	0.91	0.96	0.91	100	0.91
	LDA	0.93	0.91	0.96	0.91	100	0.91
	KNN	0.91	0.89	0.91	0.89	100	0.89
1	CART	0.92	0.91	0.94	0.91	100	0.91
	BernoulliNB	0.92	0.92	0.90	0.92	100	0.92
	SVM	0.91	0.90	0.90	0.90	100	0.90
	RF	**0.94**	**0.92**	**0.96**	**0.92**	**100**	**0.92**
	MLP	**0.94**	**0.92**	**0.96**	**0.92**	**100**	**0.92**
	LR	**0.99**	**0.99**	**0.99**	**0.99**	**74**	**0.99**
	LDA	0.98	0.98	0.98	0.98	74	0.98
	KNN	0.97	0.97	0.97	0.97	74	0.97
2	CART	0.97	0.97	0.97	0.97	74	0.97
	BernoulliNB	**0.99**	**0.99**	**0.99**	**0.99**	**74**	**0.99**
	SVM	0.97	0.97	0.97	0.97	74	0.97
	RF	0.97	0.97	0.97	0.97	74	0.97
	MLP	0.97	0.97	0.97	0.97	74	0.97

**Table 4 biomolecules-12-01552-t004:** Analysis of classification results of RF or MLP for classifier 1 and LR or BernoulliNB for classifier 2. The precision, recall score and f1-score are for each class (other variants, VUS, B/LB, P/LP) on the test set. The “Support” column indicates the total number of variants for each class based on the test set.

Classifier	Classes	Precision	Recall	Specificity	F1-Score	Support
1	Other variants	1.00	0.89	1.00	0.94	72
	VUS	0.78	1.00	0.89	0.88	28
2	B/LB	0.97	1.00	0.97	0.99	36
	P/LP	1.00	0.97	1.00	0.99	38

**Table 5 biomolecules-12-01552-t005:** Analysis of classification results of eight ML algorithms under consideration. The precision, recall score and f1-score are for the average of the three classes (P/LP, VUS, B/LB) on the test set.

ML Algorithms	Precision	Recall	Specificity	F1-Score	Support	Accuracy
LR	0.92	0.92	0.96	0.91	100	0.91
LDA	0.92	0.92	0.96	0.91	100	0.91
KNN	0.90	0.89	0.95	0.89	100	0.89
CART	0.93	0.93	0.96	0.92	100	0.92
BernoulliNB	0.90	0.90	0.95	0.89	100	0.89
SVM	0.90	0.91	0.95	0.90	100	0.90
RF	0.93	0.93	0.96	0.92	100	0.92
MLP	0.93	0.93	0.96	0.92	100	0.92

**Table 6 biomolecules-12-01552-t006:** Analysis of classification results of RF, MLP and BernoulliNB for classifier 1 and LR, BernoulliNB and LDA for classifier 2. The precision, recall score and f1-score are for the average of the classes (other variants—VUS and B/LB–P/LP) on the test set, based on the ClinVar data set.

Classifier	ML Algorithms	Precision	Recall	Specificity	F1-Score	Support	Accuracy
1	RF	0.64	0.60	0.60	0.61	11932	0.60
	MLP	0.64	0.60	0.60	0.61	11932	0.60
	BernoulliNB	0.91	0.91	0.90	0.91	11932	0.91
2	LR	0.75	0.66	0.66	0.55	8324	0.66
	BernoulliNB	0.74	0.66	0.66	0.55	8324	0.66
	LDA	0.98	0.98	0.98	0.98	8324	0.98

**Table 7 biomolecules-12-01552-t007:** Analysis of classification results of BernoulliNB for classifier 1 and LDA for classifier 2. The precision, recall score and f1-score are for each class (other variants, VUS, B/LB, P/LP) on the test set, based on the ClinVar data set. The “Support” column indicates the total number of variants for each class based on the test set.

Classifier	Classes	Precision	Recall	Specificity	F1-Score	Support
1	Other variants	0.91	0.96	0.78	0.94	8324
	VUS	0.91	0.78	0.96	0.84	3608
2	B/LB	0.95	1.00	0.97	0.97	3138
	P/LP	1.00	0.97	1.00	0.98	5186

## Data Availability

Publicly available data sets were analyzed in this study. This data and the software can be found here: https://github.com/vasilikikaral/MARGINAL-software (accessed on 10 August 2022).

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
