# Peer review of "MARGINAL: An Automatic Classification of Variants in BRCA1 and BRCA2 Genes Using a Machine Learning Model"

_biomolecules, 2022, doi:10.3390/biom12111552_

Round 1
Reviewer 1 Report
In the present work the authors attempted to create a new algorithm/approach including a machine learning model to support variant classification in BRCA1 and BRCA2 genes based on the American College of Medical Genetics and Genomics and the American Molecular Pathology (ACMG-AMP) guidelines.
Their approach is interesting and their work has merit for publication after addressing some minor issues.
The authors have introduced their concept in an interesting and complete approach. I especially liked their introduction of other methods proposed for dealing with this topic. Please add a short paragraph referring to the biological role of BRCA1 and BRCA2 in cell biology as well as why are these two genes so important in breast cancer ontogenesis?
In the "Materials and Methods" section please provide a short rationale why the three variants were chosen. Why not more variants less prevalent. Please explain your rationale.
Finally, the authors should highlight their findings at the end of the "Discussion" section in the form of conclusions. Indicate how the proposed algorithm could be used for the prognosis and/or diagnosis of breast or ovarian cancer.
Minor issues
Please connect citations (e.g. [21][22] should be [21,22])
Reviewer 2 Report
The authors present new automatic classification of variants for BRCA1 and BRCA2 genes. They proposed into two steps classifications with different ML method for each step after comparing the performance between 8 methods. The authors used a well-annotated clinical data with genomic data for the discovery phase of ML, and then tested on ClinVar data.
Major revision:
The authors did not well describe the discovery cohort and the validation cohort.
The authors should provide a table with general features of the discovery and validation cohorts based and provide a full description about the validation cohorts and if they put any criteria to select variants
The authors indicated that there is no need of IRB. Consequently they need to provide the data on supplementary table for the discovery cohort and validation cohort to allow any researchers to test the data on their models or the method proposed here and provide a more information about the version of data.
The authors need to clarify more that the discovery cohort is based on Greek cohort and can be limited about multi-ethnic origins.
The authors did not describe if they studied only variants found in cohorts and that were associated with HBOC phenotypes and if so, they need to describe the phenotypes used in their methods.
The authors provides metrics about the different methods, however, they did not provide any supplementary tables that provide the classification that each method propose for each variants and so potentially high some regions of genes that show some complexity to classify.
The authors said that they used the classification provided by CanVaS as reference. They need to provide a brief description to readers to understand why they used as a reference.
The authors present a new method and propose that this could help to improve the annotation of BRCA1 and BRAC2 variants. However, they make no reference where the readers could download and test on their data. The authors need to provide the access of their software through GitHub or Code Ocean to allow FAIR ( findability, accessibility, interoperability, and reusability).
Reviewer 3 Report
The authors proposed MARGINAL, a machine learning (ML)-based software for interpreting rare BRCA1 and BRCA2 germline variants. MARGINAL software classifies variants into three categories namely, (likely) pathogenic, of uncertain significance, and (likely) benign, implementing the criteria established by the American College of Medical Genetics and Genomics and the Association for Molecular Pathology (ACMG-AMP). There are several problems:
1. There are similar studies of activity prediction for TP53 variants (PMID: 21857971). They should be introduced and compared. In this previous study, the structure features were found to be important. But in this study, the authors did not consider structure at all. In fact, with AlphaFold2, the authors can predict the structure changes.
2. As a machine learning (ML)-based software, the authors need to upload it and the data onto GitHub.
3. The authors should clearly describe what features were used and how many features were used.
4. The authors should do feature selection and feature importance analysis.
5. In Figure 5, more data points were needed to plot the ROC curve.
Round 2
Reviewer 3 Report
The authors have made significant improvements and the revised manuscript should be accepted.
Author Response
We thank the reviewer for the comment.